# Characterization and Food Application of the Novel Lytic Phage BECP10: Specifically Recognizes the O-polysaccharide of *Escherichia coli* O157:H7

**DOI:** 10.3390/v13081469

**Published:** 2021-07-27

**Authors:** Do-Won Park, Jong-Hyun Park

**Affiliations:** Department of Food Science and Biotechnology, Gachon University, Seongnam 13120, Korea; sgm01006@naver.com

**Keywords:** *E. coli* O157:H7, bacteriophage, phage receptor, antibacterial activity, food application

## Abstract

*Escherichia coli* O157:H7 is a global concern that causes serious diseases, such as hemolytic uremic syndrome and bloody diarrhea. To control *E. coli* O157:H7 in food, a novel siphophage, BECP10, that targets the O157 serotype was isolated and characterized. Unlike other *E. coli* phages, BECP10 can only infect *E. coli* O157 strains, and thus, did not infect other strains. The 48 kbp genome of BECP10 contained 76 open reading frames (ORFs), including 33 putative functional ORFs. The phage did not contain lysogeny-related modules or toxin-associated genes, suggesting that the phage might be strictly lytic. The tail spike protein (TSP) sequence had very low homology with the reported T1-like phages, indicating that TSP might be related to this unique host spectrum. The specific O-antigen residue of *E. coli* O157:H7 may be a key factor for phage infection by adsorption and receptor identification. The phage exhibited strong antibacterial activity against *E. coli* O157:H7 over a broad pH range and showed little development of phage-insensitive mutants. The phage sustained viability on the burger patties and reduced *E. coli* O157:H7 to a non-detectable level without the emergence of resistant cells at low temperatures for five days. Therefore, phage BECP10 might be a good biocontrol agent for *E. coli* O157:H7-contaminated food matrices.

## 1. Introduction

As the number of single-person households has recently increased and food-related lifestyles have rapidly emerged, the demand for convenience foods has grown rapidly [1]. Accordingly, the market size of ready-to-eat (RTE) foods has increased worldwide, and the quality and nutritional value of RTE foods have markedly improved compared to the past [2,3]. However, due to the characteristics of RTE foods, there is always a risk of contamination by foodborne pathogens. In particular, raw materials, such as leafy vegetables, fruits, poultry meat, and beef, are the main causes of food contamination [4].

Most foodborne pathogens are problematic; however, enterohemorrhagic *Escherichia coli* (EHEC) is considered a fatal hazard in the food industry [5]. EHEC infection causes toxin-mediated damage to the intestine and renal endothelial cells, resulting in hemolytic uremic syndrome and bloody diarrhea [6]. In 2019, more than half of the EHEC outbreaks in the USA were caused by romaine lettuce and ground beef [7,8]. Many EHEC serotypes are known; however, the representative serotype is O157:H7, which is difficult to control in the food supply chain as this serotype can survive for a long time below 4 °C and is very tolerant to acidic conditions [9]. Heat processes can efficiently remove *E. coli* O157:H7; however, because these processes cannot be applied to heat-sensitive foods, alternatives such as chlorine gas, essential oil, and organic acid have been extensively studied and proposed [10]. These alternatives are known to be very effective disinfectants for controlling *E. coli* O157:H7; however, they are associated with many drawbacks, such as high cost, negative effects on food sensory features, and technical difficulties in the food supply chain [11,12]. Therefore, an effective biocontrol agent that is more manageable and does not affect the sensory properties of food is needed.

Bacteriophages (phages) are bactericidal viruses that can only target specific host bacteria and have been proposed as an attractive strategy to control foodborne pathogens [13]. Phages are not only non-toxic to humans, but also require a lower cost for large-scale production than other biocontrol agents [14]. However, the use of unverified phages can increase the emergence of bacteriophage insensitive mutants (BIMs) and drive the transduction of virulence or antibiotic resistance genes [15]. In addition, phages should be stable against various environmental stresses and exhibit strong lytic activity only for the target pathogen strain [16,17]. Therefore, phages used as biocontrol agents should be assessed in a variety of ways to prevent potential problems. Several commercial phagal products, including SalmoFresh™, ListShield™, and EcoShield™, are generally recognized as safe (GRAS) and have been used directly in the food industry [18]. *E. coli* O157:H7 infecting phages and their food application studies have also been reported. A phage cocktail of DT1 and DT6 applied to *E. coli* O157:H7 contaminated beef demonstrated a 2.58 log reduction at 24 °C [19]. Moreover, treatment of phage ECPS-6 to raw milk resulted in a 4.2 log reduction of *E. coli* O157:H7 cells despite the relatively low multiplicity of infection (MOI) [20]. However, since bacteria constantly develop a defense system against phage attacks [21], the isolation and characterization of new phages should be carried out steadily to combat foodborne pathogens.

In this study, a novel bacteriophage, BECP10 targeting *E. coli* O157:H7, was isolated and characterized. Unlike general *E. coli* infecting phages, this phage could only infect the O157 serotype *E. coli* and exhibited very strong lytic activity. To identify the BECP10 receptor, genome analysis, adsorption assays, and random mutagenesis, screening tests were performed. Ultimately, BECP10 was found to efficiently reduce *E. coli* O157:H7 in burger patties without the emergence of resistant mutants.

## 2. Materials and Methods

### 2.1. Bacterial Strains, Growth Conditions, and Primer Sets

The bacterial strains used in this study are listed in Table 1. All bacteria were grown aerobically at 37 °C for 18 h in Luria-Bertani (LB) agar and broth (Difco, Detroit, MI, USA). When needed, antibiotics were supplemented as follows (final concentrations): ampicillin (50 μg/mL) and kanamycin (50 μg/mL). The primers used in this study are listed in Appendix A.

### 2.2. Bacteriophage Isolation and Preparation

*E. coli* NCTC 12079 was used as the host bacterium for phage isolation. A sewage sample from the Seongnam Water Reclamation Center (Gyeonggi, Korea) was used as the source for phage isolation. To isolate the phage, the sewage sample was centrifuged at 8000× *g* for 10 min and the supernatant was filtered using a 0.22-μm filter system (Millipore, USA). Filtered sewage was mixed with an equal volume of 2× LB broth, and *E. coli* overnight cultures were inoculated for phage propagation. The mixture was incubated at 37 °C for 18 h. Thereafter, it was centrifuged at 8000× *g* for 10 min, and the supernatant was filtered using a 0.22-μm filter. The filtered supernatant was serially diluted using SM buffer (100 mM NaCl, 10 mM MgSO4, 50 mM Tris-HCl, pH 7.5) and added to 5 mL of LB soft agar (LB broth supplemented with 0.6% agar) with *E. coli* and overlaid on LB agar. The plates were incubated overnight at 37 °C, and a single plaque was selected and resuspended in SM buffer. The resuspended phage was then overlaid on LB agar with the host to isolate the single phage. The plaque-picking step was repeated three times. Titers of the phage were examined by the spot method based on an overlaid assay and expressed as plaque-forming units (PFU) per mL. To propagate the isolated phage, 5 mL of SM buffer was poured into a high-concentration phage plate (10^8^ PFU/mL). Thereafter, the plate was incubated for 6 h and the phage buffer was collected and filtered using a 0.22-μm filter. The collected phage lysates were stored at 4 °C for further experiments.

### 2.3. Morphological Analysis of the Isolated Phage

To concentrate the phage, phage lysate was mixed with 20% polyethylene glycol (PEG) 8000 (Sigma, St Louis, MO, USA) at a ratio of 1:1. The PEG-added phage was centrifuged at 10,000× *g*, and the phage pellet was resuspended in SM buffer. The concentrated phage (1011 PFU/mL) was washed twice using 0.1 M ammonium acetate and resuspended in SM buffer. To observe the morphology, 10 μL of phage solution was fixed on a carbon grid and negatively stained with 2% uranyl acetate. The morphology was observed using transmission electron microscopy (TEM) (H-7600, Hitachi, Tokyo, Japan) operated at 80 kV [22]. Phage was identified and classified according to the guidelines of the International Committee on Taxonomy of Viruses (ICTV) [23].

### 2.4. Phage Host Range Determination

The phage host spectrum was determined using the efficiency of plating (EOP) method, as previously described [24]. Propagated phages (1010 PFU/mL) were serially diluted 10-fold, and 10 μL of diluted phages was spotted onto the overlaid LB agar. Plaques were enumerated after 24 h of incubation at 37 °C. The EOP was determined as the ratio of the number of plaques appearing on the lawn of a test strain to the number of plaques on the reference strain, *E. coli* NCTC 12079.

### 2.5. Whole-Genome Sequencing and Bioinformatic Analysis

Phage DNA extraction was carried out using a Phage DNA Isolation Kit (Norgen Biotek Corporation, Thorold, Ontario, Canada), according to the manufacturer’s instructions. The sequencing library was prepared by random fragmentation of the DNA sample, followed by 5′ and 3′ adapter ligation [25]. Sequencing was carried out at Bioneer Inc. (Daejeon, Korea) using an Illumina NovaSeq 6000 platform. A single contig assembly was carried out using SOAPdenovo2 software [26]. After assembly, predictive open reading frames (ORFs) and possible tRNAs were predicted using Prokka v.1.14.0 [27]. The predicted ORFs were annotated using the NCBI BLASTP and InterProScan databases [28,29]. After genome annotation, the genome composition was expressed using the DNAplotter software [30]. The complete genome sequence and annotation results of *E. coli* phage vB_EcoS-BECP10 (BECP10) were deposited in the GenBank database under the accession number MW286156.1. To analyze the homologies among the T1-like phages, tblastx analysis was performed and visualized using Easyfig v.2.2.5 software with a 20% identity cut-off [31].

### 2.6. Phage Receptor Identification

#### 2.6.1. Phage Adsorption Assay

The adsorption assay was performed as previously described, with some modifications [32]. Exponentially growing *E. coli* NCTC 12079 (OD600 = 0.8–1.0) was centrifuged (8000× *g*, 10 min) and resuspended in LB broth. The cell suspension was diluted to an OD600 of 0.1–0.2 (107 CFU/mL) in 9 mL of pre-warmed LB broth. Thereafter, 1 mL of phage BECP10 was added at a multiplicity of infection (MOI) of 0.1, and incubated at 37 °C for 15 min. After 15 min of incubation, 1 mL of culture was centrifuged (15,000× *g*, 1 min) and filtered (0.22-μm filter) every 3 min to collect the unbound free phages. Unbound free phages were determined using a plaque assay.

To identify the phage receptor type, exponentially growing *E. coli* NCTC 12079 cells were treated with three solutions: sodium acetate (50 mM, pH 5.2), sodium acetate containing 100 mM periodate, and SM buffer containing proteinase K (0.2 mg/mL). Each sample was incubated at 37 °C for 2 h, and the adsorption assay was performed as described above. The baseline number was determined by adding the same concentration of BECP10 to the bacteria-free LB broth. 

#### 2.6.2. Screening of Phage BECP10 Resistant Mutants from the Tn5 Random Mutant Library

A transposon random mutant library of *E. coli* ATCC 43888 was constructed using the EZ-Tn5™ <R6Kγori/KAN-2> insertion Kit, according to the manufacturer’s instructions (Epicenter, Madison, WI, USA). A total of 1500 kanamycin-resistant mutant colonies were obtained, and each colony was cultured in 1 mL of LB broth containing 50 μg/mL kanamycin. To screen the BECP10 resistant mutants, each overnight cultured mutant was inoculated in 96-well plates containing LB broth, and BECP10 was added (MOI = 100). Plates were incubated at 37 °C for 5 h, and phage-insensitive mutants were screened by measuring the OD600 values. Screened mutants were cultured on 96-well plates in the same manner as above, and phage ECP26 (rV5-like phage) was added (MOI = 100) and lysed mutants were reselected. Genomic DNA was extracted from the reselected mutants using the AccuPrep^®^ Genomic DNA Extraction Kit (Bioneer, Korea) for rescue cloning. Rescue cloning was performed according to the EZ-Tn5™ <R6Kγori/KAN-2> insertion kit protocol. The Tn5 cassettes were transformed into *E. coli* DH5α by electroporation, and amplified Tn5 cassettes were extracted using the AccuPrep^®^ Plasmid Mini Extraction Kit (Bioneer, Korea). The transposon insertion sites were identified by Sanger sequencing (Bioneer, Korea) and BLAST analysis.

#### 2.6.3. Complementation of the Phage Insensitive Mutants

Complementation of the Tn5 cassette inserted mutants was performed by cloning the pBAD/HisA plasmid (Invitrogen, Waltham, MA, USA). The coding regions of the *rfaJ* and *wzy* genes were amplified by PCR using the primer sets wzy_O157-F, wzy_O157-R, rfaJ_O157-F, and rfaJ_O157-R, respectively. The amplified PCR product was cloned into pBAD/HisA using an Overlap Cloner DNA cloning kit (ELPIS-Biotech, Daejeon, Korea). The pBAD/HisA::*rfaJ* and pBAD/HisA::*wzy* plasmids were electroporated into *E. coli* ATCC 43888 and complementation was confirmed using the O157 latex agglutination test kit (Oxoid, Hampshire, UK).

#### 2.6.4. Phage BECP10 Sensitivity Test

Mutants and complemented strains were incubated in 5 mL of LB broth containing 1% D-glucose. When the culture reached the early exponential phase (OD_600_ = 0.5–0.6), 0.02% L-arabinose or an equal volume of 1% D-glucose was supplemented and incubated at 37°C overnight. One hundred microliters of overnight culture was added to 5 mL of LB soft agar and poured onto LB agar plates. Subsequently, 10 μL of serially diluted BECP10 and ECP26 lysates (10^9^–10^2^) were spotted on the bacterial lawns and incubated at 37 °C overnight.

### 2.7. Temperature and pH Stability of Phage

The temperature and pH stability of the phage were evaluated as previously described, with some modifications [33]. To determine the effect of temperature, phage suspensions (108 PFU/mL) were incubated at 25 °C, 37 °C, 45 °C, 55 °C, 65 °C, and 70 °C for 1 h using a PCR thermocycler (TP600, Takara, Japan). After incubation, the phages were cooled on ice for 30 min and the residual phage titer was measured through a spot method using *E. coli* NCTC 12079. To assess the pH stability, 100 μL of phage lysates was diluted in 900 μL of various premade buffers (pH 2.0; 0.1 M HCl-KCl buffer, pH 3.0 to 6.0; 0.1 M citrate buffer, pH 7.0 to 10.0) and incubated at 25 °C for 1 h. As performed in the temperature stability test, the residual phage titer was measured using the spot method after the reaction.

### 2.8. Bacterial Challenge Assay

*E. coli* O157:H7 NCTC 12079 cells were cultured until the early exponential phase in 50 mL of LB broth. At that time, BECP10 lysates were inoculated at MOIs of 100, 10, 1, and 0.1. To compare the antibacterial activity against *E. coli* O157:H7, ECP26 lysates were inoculated under the same conditions. After infection, cultures were incubated at 37 °C for 24 h, and then the growth was periodically monitored by measuring the OD_600_ using a spectrophotometer (EMC-11S-V, EMCLAB, Germany). To confirm the effect of phage on pH, the pH of the bacterial cultures was also measured every 3 h.

### 2.9. Phage Food Application 

Phage food application was conducted using a method modified from previous reports [33]. Frozen burger patties were purchased from a local market and stored at 4 °C before use. Frozen patties were separated into 25 g pieces using sterile stainless scissors. The early exponential phase *E. coli* NCTC 12079 was centrifuged at 8000× *g* for 10 min and suspended in phosphate-buffered saline (PBS, pH 7.0). The *E. coli* suspension was serially diluted (approximately 10^6^ CFU/mL) in PBS, and 1 mL was inoculated into the separated patties. After drying at room temperature, patties were treated with 1 mL of prepared phage solution (10^8^ PFU/mL in 1× SM buffer) or PBS using a reversible spray bottle (DAIHAN Scientific, Korea). Each patty was placed in a stomacher bag and incubated at 4 °C for 120 h. After the appropriate duration of time, patties were added to 225 mL of PBS containing 0.5% Triton X-100 and processed using a stomacher for 5 min. The homogenate was centrifuged at 8000× *g* for 10 min and resuspended in PBS. The survival of residual *E. coli* was determined by the plate count method using sorbitol–MacConkey agar (Oxoid, Basingstoke, Hampshire, UK), and colony counts were converted to log CFU/g. Survival of the supernatant phages was determined using the LB agar overlay assay as previously described. After incubation, plaques were counted and titers were converted to log PFU/g.

### 2.10. Statistical Analysis 

The experiments were replicated three times, and the experimental results are expressed as mean ± standard deviation (SD). All results were analyzed by one-way analysis of variance (Dunnett’s test) for a significance level of *p* < 0.05. GraphPad Prism 7.0 (GraphPad Software, San Diego, CA, USA) was used for data analysis.

## 3. Results and Discussion

### 3.1. Isolation and Characterization of Phage BECP10 

Phage BECP10 was isolated and purified from a sewage sample collected from the Seongnam Water Reclamation Center, Korea. BECP10 formed large (4.0–4.5 mm) and clear plaques on *E. coli* O157 NCTC 12079. In particular, BECP10 formed a significant halo zone around the plaque (Figure 1A). This halo is known to be formed when the cell wall components (e.g., capsular polysaccharides and exopolysaccharides) of bacteria are degraded by depolymerase in the phage, and these properties cause irreversible damage to the cell walls of target bacteria [34]. TEM revealed that BECP10 had a non-contractile tail (*n* = 6, 175 ± 10 nm) and an icosahedral head (*n* = 6, 59 ± 1 nm) (Figure 1B). BECP10 was assigned to the *Siphoviridae* family.

### 3.2. Host Range Analysis 

The host of BECP10 was determined using 10 *E. coli* O157 strains, 26 *E. coli* non-O157 strains, and other gram-positive and gram-negative bacteria (Table 1). BECP10 showed very high lytic activity against 11 *E. coli* O157 strains (EOP > 0.5), but could not infect the other tested strains, including *E. coli* non-O157. Interestingly, BECP10 did not form plaques on *Shigella* and *Salmonella* strains, in contrast to previously reported O157-infecting phages [35,36]. Notably, BECP10 could not infect the K-12 strains of *E. coli* (DH5a, BL21, and MG1655), which mostly lacked the phage defense system [37]. Phages infecting *E. coli* are known to recognize cell wall components, such as lipopolysaccharide (LPS) and channel proteins, using their tail components [38]. Therefore, BECP10 was predicted to recognize specific cell wall components of the *E. coli* O157 strains. There are several reports about the O157-specific phages such as Eco4M-7, KH1, KH4, and KH5 [39,40]. However, as with BECP10, their host ranges were determined only by plaque formation. Therefore, for further understanding of the host-phage interactions, genome sequencing or receptor analysis was needed.

### 3.3. Whole-Genome Sequencing and Bioinformatic Analysis

The BECP10 genome has a length of 47,915 bp and a GC content of 43.7%. The complete genome of BECP10 contained 76 ORFs and 1 tRNA gene (Figure 2A). Among the total 76 ORFs, 33 ORFs (43%) revealed putative functions, and similar to other general *E. coli* phages, BECP10 contained all the essential genes related to replication, packaging, and structure. BECP10 was also predicted to have a classical lysis module of *E. coli* phages composed of endolysin-holin and Rz proteins. In contrast, lysogeny-related modules (integrase, CI-Cro like repressor system, exicionase) and toxin-associated genes were not detected, suggesting that BECP10 is a strictly lytic (virulent) phage. Temperate phages are known to be unsuitable for food applications because they not only increase host immunity through lysogenic conversion, but also could transfer harmful genes (e.g., antibiotic resistance genes, Shiga toxin) [41]. On the other hand, lytic phages are known to be more effective than temperate phages as they can exclude the problems mentioned above [42], suggesting that BECP10 could be considered a safe biocontrol agent.

Annotation data revealed that the genome of BECP10 belonging to the “Jk06likevirus” genus shared most of the functional ORFs with *Escherichia* phages of vB_EcoS_AHS24, AKS96, and AHP24 (Appendix A). “Jk06likevirus” is a recently classified subfamily of T1-like virus and is known to infect only *E. coli* O157 strains; however, the receptor and receptor-binding domain (RBP) remain unknown [43]. Tail fiber or tail spike protein (TSP) is responsible for host recognition and is known to play an important role in determining the host spectrum [44]. BECP10 has two possible tail proteins (ORF34 and ORF42) that are expected to function as RBPs. Further comparative genome analysis of BECP10 and previously reported T1-like phages (AHP24 and Rtp) revealed that essential genes, such as terminase large subunit and major capsid protein, shared >70% identity at the amino acid level (Figure 2B). In addition, the three phages shared highly similar long-tail fiber proteins (82%). However, the TSP showed very low homology (<20%). Phage Rtp is known to use rough LPS for infection instead of FhuA and TonB; however, the long tail fiber is not involved in LPS recognition [45]. Therefore, the TSP might be responsible for the unique host spectrum of the “Jk06likevirus” genus.

### 3.4. Identification of the Phage BECP10 Receptor

#### 3.4.1. Phage Adsorption Assay

Based on host range analysis and comparative genomic analysis, LPS might play a crucial role in BECP10 infection. To confirm the LPS of *E. coli* O157 as a receptor for BECP10, periodate and proteinase K were used for the adsorption assay. Periodate and proteinase K are known to degrade polysaccharide and membrane protein components, respectively [32]. Without the treatments, approximately 70% of the particles in BECP10 were attached to the host cell within 3 min, and adsorption was completed within 15 min (Figure 3A). Treatment with proteinase K and acetate buffer did not affect the adsorption of BECP10, whereas treatment with 100 mM periodate significantly decreased the adsorption efficiency (Figure 3B). These results suggest that cell wall polysaccharides might be related to BECP10 infection.

#### 3.4.2. Screening of Phage Resistant Mutants and the Complementation Test 

To identify which polysaccharide component on the cell wall is involved in BECP10 infection, a Tn5 mediated mutant library of *E. coli* O157 ATCC 43888 was constructed. For a more detailed analysis, the mutants that were not infected with myophage ECP26, known to have an LPS outer core as a receptor, were excluded [33,46]. Among the 1500 mutants, two clones that were resistant to BECP10 and sensitive to ECP26 were found. The Tn5 inserted regions of mutants were sequenced and the genes, *wzy* and *rfaJ*, were identified (Figure 4A). The *wzy* gene encodes O-antigen polymerase (Wzy) and does not directly synthesize O-antigen residues; however, this gene is responsible for linking the synthesized residue [47]. The *rfaJ* gene encodes α-1,2-glucosyltransferase (RfaJ) and is involved in the synthesis of the last residue of the outer core (Figure 4B) [48]. Both mutants (*wzy*::Tn5 and *rfaJ*::Tn5) showed a negative reaction in the O157 latex agglutination test, suggesting that the O-antigen synthesis function was lost (data not shown). BECP10 could not form plaques against O-antigen mutants, but restored sensitivity against complemented strains (Figure 4C). Regardless of complementation, ECP26 could infect the host normally. These findings suggest that BECP10 might recognize LPS as a receptor, but specifically recognize the O-antigen itself, not the core polysaccharide. O-antigen (smooth LPS)-specific phages are known to recognize and degrade the O-antigen using the tail fiber or TSP at the beginning of infection, and podoviruses, such as P22, LKA1, and PHB19, are best characterized [49,50,51]. Based on both previous results and mutagenesis tests, BECP10 recognizes the specific O-antigen residues of *E. coli* O157 strain using its TSP.

### 3.5. Temperature and pH Stability of Phage

To evaluate the temperature and pH stability of BECP10, phage lysates were exposed to various temperatures and pH levels for 1 h (Figure 5). The phage was very stable at 25–45 °C. However, there was a significant reduction in the titer when exposed to 55 °C and 65 °C (1.5 and 5 log PFU/mL, respectively). Further, at 70 °C, the phage was completely inactivated. The phage also formed plaques normally without losing its activity in the pH range of 4–9. In contrast, a slight titer decrease was observed in the pH 3 and pH 10 solutions (1.4 and 0.6 log PFU/mL, respectively), while complete inactivation occurred in the pH 2.0 solution. Temperature and pH are important factors in terms of storage and application in food [16]. Although BECP10 was very unstable above 65 °C, similar to previously reported *E. coli* infecting phages, it showed relatively high stability over a wide pH range. Therefore, BECP10 could be applied flexibly in various food environments.

### 3.6. Bacterial Challenge Assay

To evaluate the effectiveness of BECP10, a bacterial challenge assay was performed against *E. coli* O157:H7. After control infection of *E. coli* with ECP26 (MOI: 10), the host bacterial growth was found to be inhibited after 1 h and this inhibition lasted for 10 h (Figure 6A). The inhibitory activity was also similar in the MOI 10 treatment group. However, treatment with a low concentration of phage (MOI: 0.1, 0.01) did not markedly inhibit bacterial growth for 24 h. For the MOI 10 and 1 groups, BIMs were observed despite ECP26 being a lytic phage. 

In contrast to phage ECP26, regardless of the MOI, BECP10 showed a strong and consistent inhibitory effect on *E. coli* and did not cause BIMs for 10 h (Figure 6B). Although the high phage concentration (MOI: 10) shortened the killing latency of the phage, the inhibition efficiency was not necessarily proportional to the MOI, as observed for MOIs 1 and 0.1. Moreover, at the end of culture, the emergence of BIMs was observed at low MOIs (0.01 and 0.1) but not at high MOIs (1 and 10). These results were similar to those of recently reported *E. coli* O157:H7 infecting myophage KFS-EC; however, KFS-EC caused the BIMs, even at a relatively high MOI of 100, unlike BECP10 [52]. In addition, both phage ECP26 and BECP10 did not significantly affect the pH of bacterial culture (Appendix A). Therefore, as the emergence of BIMs is one of the biggest hurdles in phage applications, the killing activity of BECP10 against *E. coli* O157:H7 could be a good advantage.

### 3.7. Phage Food Application

To validate the biocontrol effects of BECP10 against *E. coli* O157:H7-contaminated food, we artificially contaminated burger patties with *E. coli* O157:H7 and sprayed PBS buffer or BECP10 (MOI: 100) onto the patties. After the patties were incubated at 4 °C, the survival of *E. coli* and phages was counted every 24 h (Figure 7). During storage at 4 °C, *E. coli* in PBS-treated patties did not decrease for 120 h. In contrast, *E. coli* in the phage-treated patties was significantly reduced by 1.8 log CFU/g within 48 h (*p* < 0.01). Thereafter, the survival rate was found to steadily decrease. The cells were reduced by 2.2 log CFU/g (*p* < 0.001) in 96 h and colony counting was impossible at 120 h. The phage populations maintained their initial concentration for 120 h (approximately 7.8 log PFU/g), suggesting that the phage inactivated *E. coli* O157:H7 using the “lysis from without (LO)” mechanism. LO is usually mediated by virion or tail-associated lysin, and its activity is known to be proportional to the phage titer and does not produce progeny [53]. Several food application experiments using the LO mechanism have been reported in previous studies. The phage cocktail product, EcoShield™, reduced *E. coli* O157:H7 cells in beef steak by 98% at refrigerated temperatures within 7 days [54]. Phage BPECO19 was also applied against *E. coli* O157:H7 contaminated raw beef at 4 °C, and was found to significantly reduce the viability of *E. coli* O157:H7 (approximately 5.2 log CFU/cm^2^) within 120 h at a high MOI (100,000) [55]. In this study, BECP10 sustained lytic activity with a relatively low titer and could reduce *E. coli* O157:H7 in the food matrix to a non-detectable level without the emergence of resistant cells at low temperatures. These promising properties are great advantages for food applications.

## 4. Conclusions

Food contamination of *E. coli* O157:H7 has been recognized as a global issue that causes fatal diseases and mainly occurs in undercooked materials. In this study, we isolated and characterized the novel *E. coli* O157-targeting lytic phage, BECP10. Herein, BECP10 was suggested to recognize a specific O-antigen residue of *E. coli* O157, which might be responsible for its unique host spectrum. BECP10 has properties that are suitable for food applications and can efficiently control *E. coli* O157:H7 even at refrigerated temperatures. However, research on the TSP of “Jk06likevirus” genus, which belongs to phage BECP10, has not been carried out. Further studies on the TSP of BECP10 are necessary for better phage application.

## Figures and Tables

**Figure 1 viruses-13-01469-f001:**
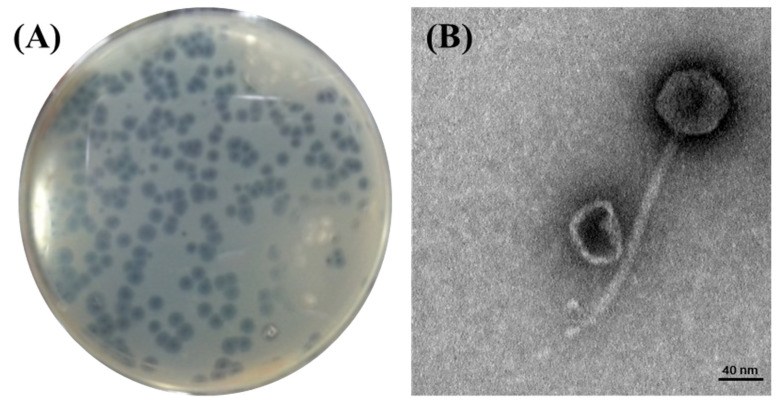
Plaque and TEM morphology of the phage, BECP10. (**A**) Plaques formed on *E. coli* NCTC 12079 by phage BECP10. (**B**) TEM morphology of phage BECP10.

**Figure 2 viruses-13-01469-f002:**
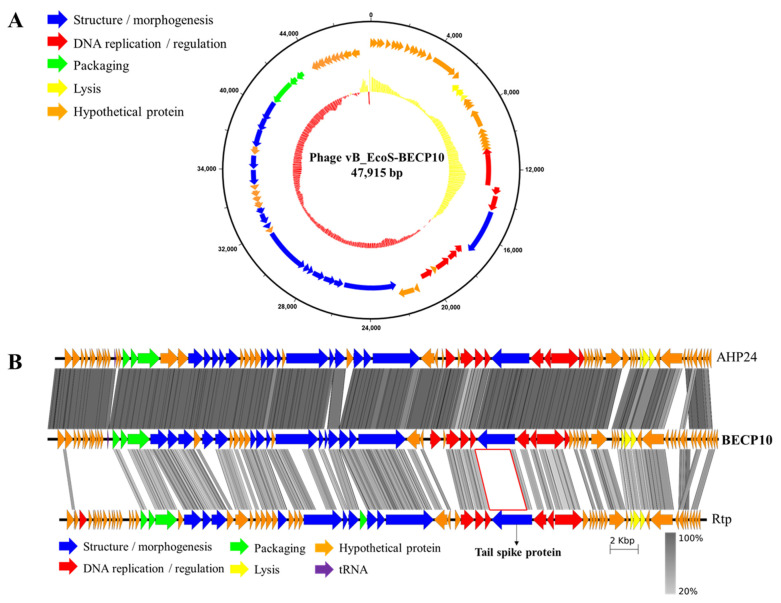
Genome map and comparative genomic analysis of phage BECP10. (**A**) On the inner ring, percentage GC is shown relative to the mean GC content of the genome. A high GC content is indicated in red and a low GC content is indicated in yellow. Outer circle indicates predicted coding DNA sequences (CDSs). (**B**) Comparative genomic analysis of the phages AHP24, BECP10, and Rtp, using the Easyfig tBlastX mode. Coding regions are represented by colored arrows. The gray-scaled shaded area indicates the different levels of amino acid similarity among the phage sequences. Scale unit means base pair. In both figures, the color of each gene indicates the functional category; categories are shown on the left or below the map.

**Figure 3 viruses-13-01469-f003:**
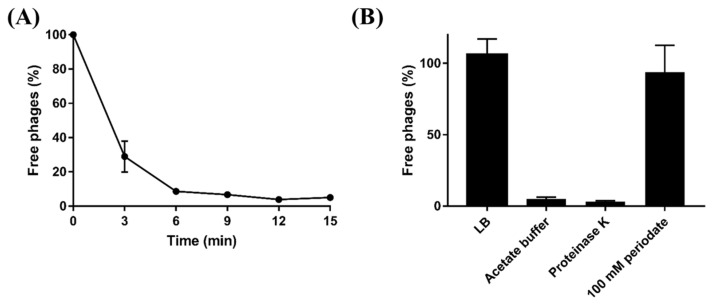
Adsorption assay of BECP10. (**A**) BECP10 adsorption assay against *E. coli* NCTC 12079 at 37 °C for 15 min. (**B**) Effect of periodate and proteinase K treatments on the adsorption efficiency of BECP10. The LB broth in BECP10 was considered as 100% (control) and the free phage rate was determined after incubation for 15 min at 37 °C.

**Figure 4 viruses-13-01469-f004:**
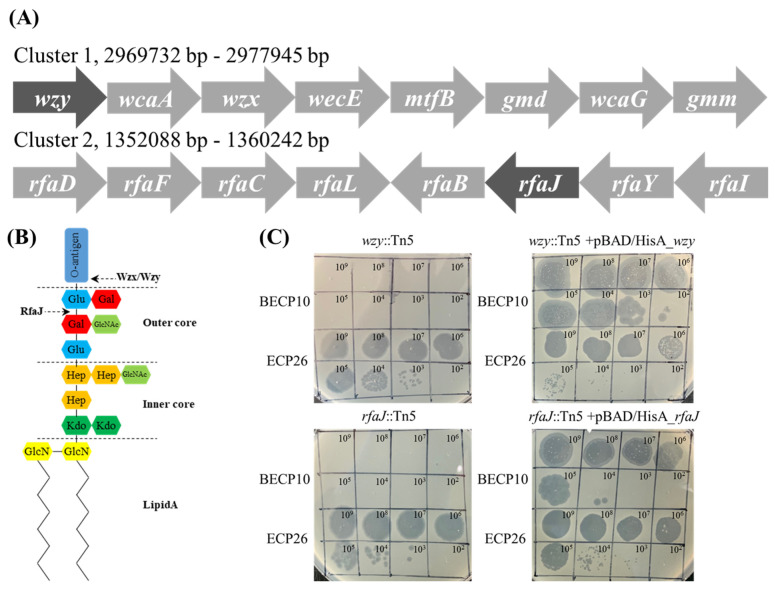
Identification of the BECP10 receptor using Tn5 random mutagenesis and complementation test. (**A**) Phages BECP10 and ECP26 were spotted on *E. coli* ATCC 43888 *wzy*::Tn5 mutant, *rfaJ*::Tn5 mutant, and their complementation strain (complemented by pBAD/HisA). (**B**) Composition of the O-antigen synthesis cluster (Cluster 1) and LPS core synthesis (Cluster 2). Dark shade means the Tn5 inserted gene. (**C**) Schematic of *E. coli* O157 LPS structure. RfaJ and Wzx/Wzy involved in outer core assembly and O-antigen synthesis, respectively.

**Figure 5 viruses-13-01469-f005:**
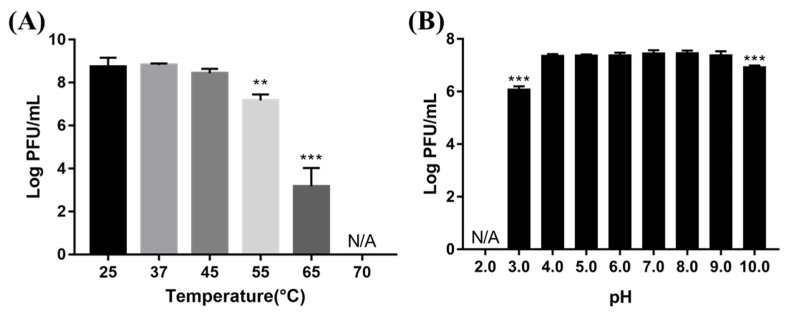
Effect of various temperatures (**A**) and pH conditions (**B**) on phage viability. All experiments were independently performed in triplicate, and the results are presented as mean ± SD. Asterisks indicate statistical significance (** *p* < 0.01, *** *p* <0.001). N/A: not available.

**Figure 6 viruses-13-01469-f006:**
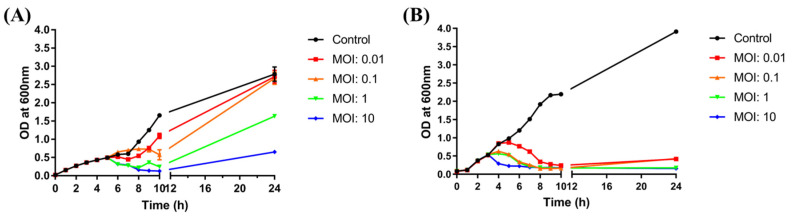
Bacterial challenge assay with ECP26 and BECP10. (**A**) Antibacterial activity of phage ECP26 according to various MOIs at 37 °C. (**B**) Antibacterial activity of phage BECP10 according to various MOIs at 37 °C. Phages were inoculated at 3 h after bacterial growth and the OD_600_ was periodically monitored to determine phage activity for 24 h. SM buffer was used as the negative control.

**Figure 7 viruses-13-01469-f007:**
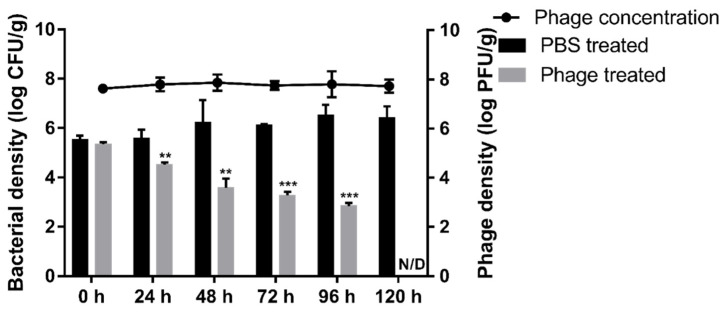
Food application of phage BECP10. The biocontrol effect of BECP10 against *E. coli* O157:H7 contaminated burger patties was evaluated. Line and bar graphs represent the viability of phage and *E. coli* O157:H7 cells on patties, respectively. The dotted line represents the limit of detection for the viable cell count. All experiments were independently performed in triplicate, and the results are presented as mean ± SD. Asterisks indicate statistical significance (** *p* < 0.01, *** *p* < 0.001). N/D: not detected.

**Table 1 viruses-13-01469-t001:** Host spectrum of the phage, BECP10.

Bacterial Strains	O-Antigen Serotype	Source or References	BECP10
*Escherichia coli* O157:H7			+
14028	O157 antigen	NCCP	+
13899	O157 antigen	NCCP	+
13930	O157 antigen	NCCP	+
13939	O157 antigen	NCCP	+
12079	O157 antigen	NCTC	+
43888	O157 antigen	ATCC	+
505B	O157 antigen	LAB collection	+
204P	O157 antigen	LAB collection	+
30-2C4	O157 antigen	LAB collection	+
W2-2	O157 antigen	LAB collection	+
***Escherichia coli* Non-O157:H7**			
25922	O6 antigen	ATCC	−
12014	O55 antigen	ATCC	−
15597	Smooth LPS	ATCC	−
15959	O5 antigen	NCCP	−
15962	O21 antigen	NCCP	−
15958	O22 antigen	NCCP	−
13919	O26 antigen	NCCP	−
13921	O26 antigen	NCCP	−
13893	O26 antigen	NCCP	−
14018	O26 antigen	NCCP	−
13937	O26 antigen	NCCP	−
13987	O55 antigen	NCCP	−
13927	O55 antigen	NCCP	−
15960	O55 antigen	NCCP	−
13988	O55 antigen	NCCP	−
13999	O55 antigen	NCCP	−
15956	O103 antigen	NCCP	−
13979	O104 antigen	NCCP	−
14540	O111 antigen	NCCP	−
15955	O113 antgien	NCCP	−
15954	O145 antigen	NCCP	−
14010	O159 antigen	NCCP	−
15953	O174 antigen	NCCP	−
DH5α	Rough LPS	Invitrogen	−
MG1655	Rough LPS	LAB collection	−
BL21(DE3)	Rough LPS	LAB collection	−
**Other Gram-negative bacteria**			
*Salmonella enteritidis* 12021	Smooth LPS	KCCM	−
*Salmonella* Typhimurium 14028	Smooth LPS	ATCC	−
*Salmonella* Typhimurium 13311	Smooth LPS	ATCC	−
*Shigella flexneri* 29903	O2 antigen	ATCC	−
*Shigella boydii* 8700	O2 antigen	ATCC	−
*Vibrio parahaemolyticus* 17802	O1 antigen	ATCC	−
*Cronobacter sakazakii* 29544	O1 antigen	ATCC	−
**Other Gram-positive bacteria**			
*Staphylococcus aureus* 6538	-	ATCC	−
*Bacillus cereus* 14925	-	KCCM	−

(1) Symbols: +, clear plaque (EOP > 0.5); −, insensitive to phage BECP10. (2) ATCC, American Type Culture Collection; KCCM, Korean Culture Center of Microorganisms; NCTC, National Collection of Type Cultures; NCCP, National Culture Collection for Pathogens.

## Data Availability

All the data presented in this study are available within the article.

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
