# Peer review of "Characterization and Food Application of the Novel Lytic Phage BECP10: Specifically Recognizes the O-polysaccharide of Escherichia coli O157:H7"

_viruses, 2021, doi:10.3390/v13081469_

Round 1

Reviewer 1 Report

First at all, the authors order is different in susy system and in main text. also, some 0157:H7 specific bacteriophages also were presented [https://journals.asm.org/doi/full/10.1128/AEM.65.9.3767-3773.1999]

in the introduction, authors should indicate where the bacteriophages against E.coli o157:h7 were used and poses some activity some examples: RTE (http://www.medycynawet.edu.pl/images/stories/pdf/pdf2017/072017/201707422424.pdf), milk (https://onlinelibrary.wiley.com/doi/abs/10.1111/jfs.12747), raw meat  (https://journals.asm.org/doi/full/10.1128/AEM.05493-11), meat products (https://www.frontiersin.org/articles/10.3389/fcimb.2013.00020/full). also bacteriophages could be used as a some kind of sanitizers for food production processing (10.1111/j.1365-2672.2005.02659.x) or biocontrol in animal farming (https://asp.zut.edu.pl/2015/14_1/asp-2015-14-1-297.pdf).

Author Response

Reviewer 1

Comments and Suggestions 1-1

First at all, the authors order is different in susy system and in main text. also, some O157:H7 specific bacteriophages also were presented.

[https://journals.asm.org/doi/full/10.1128/AEM.65.9.3767-3773.1999]

<Answers to 1-1>

We thank the reviewer for this suggestion. To avoid distractions, we added the appropriate sentences including two references (including the references suggested by reviewers) in Results and Discussion as followed. “There were several reports about the O157-specific phages such as Eco4M-7, KH1, KH4, and KH5 [37, 38].” (Lines 250-254) Also, to avoid confusion, the sentence "unlike previously reported E. coli-infecting phages" in line 66 was changed to "unlike general E. coli infecting phages".

Above mentioned phages (Eco4M-7, KH1, KH4, and KH5) were demonstrated only by the host spectrum tests based on the plaque assay. Therefore, to emphasize the necessity for further studies, the sentences were in Results and Discussion as followed. “However, as with BECP10, their host ranges were determined only by plaque formation. Therefore, for further understanding of the host-phage interactions, genome sequencing or receptor analysis was needed.”

Comments and Suggestions 1-2

In the introduction, authors should indicate where the bacteriophages against E.coli O157:H7 were used and poses some activity some examples:

RTE(http://www.medycynawet.edu.pl/images/stories/pdf/pdf2017/072017/201707422424.pdf), milk (https://onlinelibrary.wiley.com/doi/abs/10.1111/jfs.12747), raw meat (https://journals.asm.org/doi/full/10.1128/AEM.05493-11), meat products (https://www.frontiersin.org/articles/10.3389/fcimb.2013.00020/full).

Also bacteriophages could be used as a some kind of sanitizers for food production processing (10.1111/j.1365-2672.2005.02659.x) or biocontrol in animal farming (https://asp.zut.edu.pl/2015/14_1/asp-2015-14-1-297.pdf).

<Answers to 1-2>

We appreciate this reviewer’s valid suggestion. In the introduction, the other research results about food application studies were added, including also the references suggested by the reviewer. E. coli O157:H7-infecting phages and their food application studies have also been reported. Phage cocktail of DT1 and DT6 applied to the pathogen-contaminated beef that demonstrated reduction of 2.58 log cells at 24°C [19]. Moreover, treatment of phage ECPS-6 to raw milk resulted in 4.2 log reduction of E. coli O157:H7 cells despite of the relatively low multiplicity of infection (MOI) [20].

Reviewer 2 Report

The authors studied the characteristics of phage BECP10 and tested its inhibition effects on E. coli O157:H7 in food. They presented some interesting findings. However, the experimental design should be improved and more data should be provided.

Line 182: Why were these temperatures selected? The food application was tested at 4 degree C, but this temperature was not included.

Since temperature and pH were tested, you should also provide information about how pH changed over time in LB and frozen burger patties for Bacterial challenge assay and Phage food application, respectively.

Author Response

Reviewer 2

The authors studied the characteristics of phage BECP10 and tested its inhibition effects on E. coli O157:H7 in food. They presented some interesting findings. However, the experimental design should be improved and more data should be provided.

Comments and Suggestions 2-1

Line 182: Why were these temperatures selected? The food application was tested at 4 degree C, but this temperature was not included.

<Answers to 2-1>

We appreciate sincerely this constructive comment. Temperature is known to be a crucial factor for bacteriophage survivability. In the food industry, commercialization is impossible unless the temperature stability of phage is verified. Therefore, we performed the temperature stability at various temperatures (25°C~70°C).

In case of 4°C stability test, because phages have been known to be highly stable in 4°C, we did not include the data about stability test in 4°C. Tailed phages were the most resistant to storage and showed the longest survivability; some of them retained viability even after 10–12 years at 4°C (Ackermann et al., 2004). Nonetheless, for this revision, since we also agree with the reviewer's suggestion, we measured the viability of phage BECP10 (5 month storaged phage) at 4°C in the our Laboratory. We confirmed the phage BECP10 still retained almost the same stability to the beginning titer of 9 log PFU/mL. Therefore, the phage BECP10 would be stable at 4°C and could be applied in food industry without the titer reduction problem.

Comments and Suggestions 2-2

Since temperature and pH were tested, you should also provide information about how pH changed over time in LB and frozen burger patties for Bacterial challenge assay and Phage food application, respectively.

<Answers to 2-2>

We appreciate this valid suggestion. Like temperature, pH is one of the important factors affecting phage survivability. Therefore, we examined the pH stability of the phage and BECP10 maintained the titer similarly in a very wide ranges of pH. It was confirmed that pH had little effect on the titer of BECP10, so we conducted the challenge assay and food application without pH measurement. However, since the reviewer's suggestion was also very reasonable, the bacterial challenge assay was repeated for this defense, and pH was measured every 3 h for 24 h. Regardless of the MOI, both the phage ECP26 and BECP10 treatment groups maintained pH 6.9-7.0 in LB (data in Suplementary Fig. 1), and there was no meaningful difference with the control group. Ususally, pH change might be detected in the liquid of LB than in the solid of patties.

Therefore, it is assumed that BECP10 will not change the pH even in food applications. The pH changes during the challenge assays are presented in the Supl. Figure 1, and the related descriptions were added in Lines 143-151, Lines 201-202 and Lines 369-370 of the revised manuscript.

Round 2

Reviewer 1 Report

the manuscript has been sufficiently improved by authors

Reviewer 2 Report

N.A.